# Actionable tests and treatments for patients with gastrointestinal cancers and historically short median survival times

Howard W. Bruckner[1]*, Fred Bassali[1], Elisheva Dusowitz[1], Daniel Gurell[2], Abe Book[1], Robert De Jager[1]

**1** MZB Foundation for Cancer Research, New York, NY, United States of America, **2** Department of Diagnostic Radiology, University Diagnostic Imaging, Bronx, New York, United States of America

* howardbruckner@gmail.com

**Data Availability Statement:** All relevant data are within the paper and its Supporting Information files.

## Abstract

### Background

Patients have difficult unmet needs when standard chemotherapy produces a median survival of less than 1 year or many patients will experience severe toxicities. Blood tests can predict their survival.

### Methods

Analyses evaluate predictive blood tests to identify patients who often survive 1 and 2 years. A four-test model includes: albumin, absolute neutrophil count, neutrophil-lymphocyte ratio, and lymphocyte-monocyte ratio. Individual tests include: alkaline phosphatase, lymphocytes, white blood count, platelet count, and hemoglobin. Eligible patients have advanced: resistant 3rd line colorectal, and both resistant and new pancreatic and intrahepatic bile duct cancers. Eligibility characteristics include: biopsy-proven, measurable metastatic disease, NCI grade 0–2 blood tests, Karnofsky Score 100–50, and any adult age. Drugs are given at 1/4–1/3 of their standard dosages biweekly: gemcitabine, irinotecan, fluorouracil, leucovorin, and day 2 oxaliplatin every 2 weeks. In case of progression, Docetaxel is added (except colon cancer), with or without Mitomycin C, and next cetuximab (except pancreatic and KRAS BRAF mutation cancers). Bevacizumab is substituted for cetuximab in case of another progression or ineligibility. Consent was written and conforms with Helsinki, IRB, and FDA criteria (FDA #119005).

### Results

Median survival is 14.5 months. Of 205 patients, 60% survive 12, and 37% survive 24 months (95% CI ± 8%). Survival is > 24, 13, and 3.8 months for patients with 0, 1–2, and 3–4 unfavorable tests, respectively. Individual "favorable and unfavorable" tests predict long and short survival. Neither age nor prior therapy discernibly affects survival. Net rates of clinically significant toxicities are less than 5%.

**Funding:** The Marcus Foundation, MZB Foundation for Cancer Research, and Aid L'Shalom Foundation The MZB Foundation for Cancer Research provided and supervised the staff, design, and analysis of this study. The other funders had no role in study design, data collection and analysis, decision to publish, or preparation of the manuscript.

**Competing interests:** The authors have declared that no competing interests exist.

## Conclusion

Treatments reproduce predictable, greater than 12 and 24-month chances of survival for the aged and for patients with drug-resistant tumors. Evaluation of blood tests may change practice, expand eligibility, and personalize treatments. Findings support investigation of drug combinations and novel dosages to reverse resistance and improve safety.

## Introduction

The sequence of treatments was designed in order to safely produce median and 24 months survival of $> 12$ and 33%, respectively, for patients with unmet needs due to their tumors' *de novo* and acquired resistance to drugs, historically short median survival times (MSTs), or poor tolerance to standard dosages of chemotherapy. The reproducible unusually long MST of the patients prompts investigation of predictive blood tests (PBTs) [1–3].

Evaluations of patients treated with the test regimens for advanced pancreatic cancer (APC), intrahepatic bile duct (CCA), each with or without prior therapy, and $\geq 3^{rd}$ line resistant colorectal cancer (RCRC) corroborate that the combinations can meet these objectives [3]. Historically, the rates of treatment-limiting adverse events (AEs) of hundreds of similar patients with expected or prior poor tolerance to standard treatment is 5% [1, 2].

In effect, our experience finds recombination and serial re-challenge with 4–6 first- and second-line standard drugs, with or without supplementary drugs, can be effective and unusually safe when all of the cytotoxins are electively administered simultaneously at 1/4–1/3 of their standard dosages. Survival is unexpectedly long, $> 12$ months, even after a tumor was demonstrably resistant during active treatment with the standard dosages of these same drugs [3].

Historical standard treatments produce an anticipated MST of 8–10 months for patients and at best 13 months for patients with no prior therapy for CCA. With standard treatment, the expected 2-year survival for each group of patients is 5–15% [4–9].

Currently, many of these patients may be undertreated due to the dire expectations of poor survival or their anticipated high rates of severe AEs due to prior treatment, age, and frailty [10, 11]. At least half of all the patients with advanced GI cancers are elderly. Standard 20–40% reduction of dosage may not avoid the, ~ 20% each, rates of severe gastrointestinal, neurological, or other limiting AEs associated with current chemotherapy [4, 7, 9, 12].

An estimated 1.1 million people were diagnosed with GI cancers in 2020. One hundred thousand deaths alone were due to GI, Colorectal, pancreatic, bile duct, and gastric cancers in the US. More than ten times that number occurred worldwide [10, 11, 13, 14]. These high mortality rates identify patients whose diagnoses create unmet needs.

The rationale for the design of the regimens is based on novel laboratory criteria and clinically validated drug interactions. To be included in the combination, drugs in pairs must have; optimum inhibitory concentrations (IC), at 12 (6–25) % of each drugs' optimum IC, reverse the *in vitro* resistance of tumors to 3 of the drugs in the combination, and reverse the resistance of $> 1$ other epithelial (gastrointestinal, gynecologic, urologic, lung, or breast) tumors to the same drugs [1, 2]. In contrast to the dosages selected for the combination, the same two drugs can be antagonistic at their IC 50.

These infrequently evaluated strategies, re-challenge in the form of four drugs, GFLIO, with or without prior response to the same drugs, can produce 6 synergistic drug interactions. When the drugs are recombined or supplemented with the addition of 1–2 new drugs to the

core regimen, each drug produces $\geq 3$ new drug synergistic interactions. The 6–7 drugs produce 6–10 simultaneously available synergistic drug pairs.

GFLIO-like regimens can also improve many patients' tests of immune function, increase exposure to RCRC's neoantigens, and produce high rates of response and 5-year survivors [15, 16]. Patients with other diseases, HIV, hypertension, hematological tumors, and solid tumors also benefit from other mechanisms attributable to multi-drug therapy [17–19].

Retrospective analyses with validated models, A.L.A.N. scores (AS), and also individual PBTs can identify retrospective groups of patients with "favorable" and "unfavorable," long and short survival [3, 20–29]. The AS, constituted by adding the number of individual unfavorable tests can predict the survival chances of long, intermediate, and short, 20, 12, and 6 months, for patients with primary CCA [27, 29]. Other models, a set of individual blood tests as part of the investigation of advanced gastric cancers, can identify groups of patients with sufficient survival to allow their inclusion in trials in spite of a performance status (PS) of 2–3 and can sometimes identify false-positive and false-negative trials [20, 21]. The neutrophil-lymphocyte ratio (NLR) can be used to compare treatments for patients with resistant APC (RAPC) [6]; however, there has been very little similar investigation of PBTs for contemporary heavily treated patients with advanced tumors that are resistant to multi-drug regimens. Historically, similar patients with "favorable" AS scores of 1–2 and low NLRs do not have an MST of 12 months [5, 6, 26, 27].

Leading investigators have proposed a phase 2 MST of 12-months as a criterion for the development of new regimens for patients with APC. They also recognize a need to identify prognostic biomarkers [22]. PBTs may be surrogate biomarkers.

## Methods

Kaplan-Meier, Cox, log-rank, and Greenwood's analyses with 95% confidence intervals (CI) examine intent-to-treat survival from first dose of gemcitabine as part of treatment with GFLIO (Table 1). This analysis was conducted in an oncology-only outpatient clinic. The objectives of the current analyses include identification of blood tests that can: compare patients and regimens, support the survival findings, and contribute to the design of further investigation. The trial was expanded with IRB approvals because of the compassionate unmet needs, objective of more information, and identification of the characteristics of patients that will gain benefit.

In the analyses, the patients are divided into "favorable relatively long, and unfavorable relatively short" survivors. Groups are split with respect to: AS and individualized elements of the

**Table 1. Sequential treatment for advanced gastrointestinal cancers.**

| Line 1 in order (GFLIO) | Line 2 in order (DM added) | Line 3 in order (Added only One) |
|---|---|---|
| Gemcitabine 500 (-20%) mg/M$^2$ 50 min | Gemcitabine 400 mg/M$^2$ 40 min | Cetuximab 400/200 mg/M$^2$ As first drug |
| Leucovorin 180 mg/M$^2$ | Leucovorin 180 mg/M$^2$ | Bevacizumab 10 mg/kg Line 3 or 4 as first drug |
| Irinotecan 80 (-25%) mg/M$^2$ 90 min | Irinotecan 60 mg/M$^2$ | — |
| Fluorouracil 1200 mg/M$^2$ 24 hrs | Fluorouracil 1200 mg/M$^2$ | — |
| Day 2, hr-20 | Day 2, hr-18 | — |
| — | Docetaxel* 25(-20*,40%) mg/M$^2$ | — |
| Oxaliplatin* 40 (-25%) mg/M$^2$ | Oxaliplatin 30 mg/M$^2$ | — |
| All q2weeks | Mitomycin* 6 (-33%) mg/M$^2$ | — |

complete blood count (CBC) and complete metabolic panel (CMP) tests [23–25, 27–29]. These examine the validated AS and reported effective PBTs' relationship to overall survival with new clinical characteristics such as age, prior therapy (resistance), and gender. The reported cut-off criteria are employed in preference to a statistical optimum cut-off for this group of patients.

The individual patients' AS, 0, 1–2, and 3–4, is calculated by adding the number of tests that predict short survival, i.e., serum albumin < 3.5 g/dL, NLR > 3.0, absolute neutrophil count (ANC) > 8,000/mm$^3$, and lymphocyte-monocyte ratio (LMR) < 2.1 [27, 29]. Historically, PBTs that predict long survival include but are not limited to lymphocytes > 1.500/µl, platelets < 300,000/µl, ANC < 8,000/mm$^3$, serum albumin > 3.5 g/dL, alkaline phosphates < 100 IU/L, and cell ratios NLR < 3.0 and LMR > 2.1 [23–25, 27–29].

Eligible patients are: adults with Karnofsky scores of 100–50 with National Cancer Institute (NCI) grade 0–2 blood tests [30]. (The objective was to use eligibility criteria that are easily accepted for future cooperative group trials.) Patients have measurable, advanced, stage IV tumors: ductal pancreatic cancers ± prior standard therapy; intrahepatic bile duct cancer ± prior standard therapy, or colorectal cancers that have failed two or more lines of standard therapy. Other requirements include: preregistration, intent-to-treat, real-time independent review of safety, Helsinki practice, written consent, and biopsy [31]. Expected survival is > 6 weeks with the MST < 10 months. The reason to omit treatment from Naïve patients are: included prior independent consultation at cancer centers with the recommendation that standard treatment would be ineffective or not worth the risk. The risks include: age, co-conditions, tumor characteristics, extent, volume, number, and poor organ functions. Ineligible patients have: central nervous system metastases, current intravenous or hospital care, unpredictable ability to reach the office, or grade 3–4, NCI 2.1, chronic-limiting AEs [30]. Failure of prior therapy or resistance of the tumor to standard drugs is defined as the discovery of new or enlarged measurable tumor during active treatment [1–3].

The treatment consists of GFLIO (Table 1), bi-weekly gemcitabine 500 mg/M$^2$, 5-fluorouracil 1200 mg/M$^2$ over 24 hours, leucovorin 250 mg in total, irinotecan 80 mg/M$^2$, and on day 2, oxaliplatin 40 mg/M$^2$. Then, at the time of progression, docetaxel 25 mg/M$^2$—and only for patients with good marrow reserve, Mitomycin C 6 mg/M$^2$ are added without discontinuation of the GFLIO drugs. Cetuximab is added as a weekly treatment in the case of further progression. In case of ineligibility for treatment with cetuximab, it is replaced by bevacizumab 10 mg/kg as a biweekly treatment. Docetaxel is omitted from the treatment of patients with colorectal cancer. Cetuximab is omitted from the treatment of patients with pancreatic cancers and other tumors with RAS/BRAF mutations. This work preceded the availability of checkpoint drugs or approved treatment for tumors with RAS/BRAF, HER-2, or BRCA mutations.

Reports describe methods for monitoring, dose modification, and support measures [1, 2]. Treatment without dose escalation may produce brief intermittent neutropenia, 1500–800, or thrombocytopenia, 125,000–80,000/µl. There is no initial use of granulocyte colony-stimulating factor (G-CSF); G-CSF 300 mcg is used for two days when needed, either day 7 for ANC near 1000/µl, projected to produce gr 4 neutropenia or day 15, ANC of < 1250/µl in order to produce an ANC of > 1250/µl to allow treatment on day 17, with no changes in dosages of chemotherapy. G-CSF is given day 7, thereafter, and the number of days is halved compared to prior treatment if the ANC on day 15 is > 8000/µl.

Initial dosages are reduced, as shown (*%) in Table 1, for prior: AEs that limit treatment due to drugs, ANCs < 1000/µl, platelet counts of < 90,000/µl, or a > 7-day delay of treatment [1–3].

Re-escalation in ½ steps: when a response ends, is not to exceed initial level 1 dosages. Escalation in half steps also occurs after an initial omission or dose reduction. Initial omission of

oxaliplatin, or omission of Mitomycin C and reduction of the docetaxel (to a starting dosage of 15 mg/M$^2$), if there are concerns due to the patients' frailty, poor PS, or prior: gr 4 ANC or platelet count nadirs; slow recovery of counts; delay of treatment > 7 days; sepsis, or inability to tolerate sepsis. The drug can be introduced in cycle 3 at the dosages shown in parentheses in Table 1. The dosage is increased stepwise as above. "Stop-go" practices hold treatment with bevacizumab and irinotecan until the complete resolution of gr 1–2 enteritis or stomatitis. Fluorouracil is escalated monthly (cycles 3 and 5), to 1400 and then 1600 mg/M$^2$, in the absence of stomatitis or enteritis, and reduced by 200 mg/M$^2$ when needed to prevent their recurrence.

Entry started in 5/2016 and closed 5/2018. The last data analyzed was entered 5/1/2019. This study was conducted under an application from the U.S. Food and Drug Administration (FDA) (# 119005). The Western, NY Downtown, Cabrini, St. Vincent, and Lutheran Hospital IRBs approved of this study through written consent. IRB requirements and support included real-time assessment of less than 5% rates of AEs and 6-month survival, and clinical benefits for > 50% of patients.

## Results

Patients include: 53 with resistant APC; 50 with resistant CRC; 19 with resistant CCA; 53 with new, no prior treatment APC; and 16 with new, no prior treatment CCA. The combined group consists of 205 patients, 122 (60%) with resistant cancers and 83 (40%) without prior chemotherapy (Fig 1).

The overall MST is 14.5 months. Sixty percent survive 12, and 37% survive 24 months (95% CI ±8%). Patients with CCA, RCCA, RCRC, and APC have an MST of > 12 months (Fig 2). The MST of patients with RAPC is 9.3 months (CI 6.3–14.3 months), 44% survive ≥ 12 months, and 20% survive ≥ 2 years.

### Age

Survival for groups of young and old patients is similar (HR = 1.06, P = 0.77) (Fig 3, Table 2). There are no significant differences in survival when the ages are set at <, or ≥ 60, 65, 70, or 75 years. In exploratory analyses, advanced age is associated with small changes in the frequency of each PBT. The rate of some tests that predict short survival increases but others decrease with age.

### Gender

Females have better survival than males. The HR is 0.71 and the P value is 0.05 (Fig 4, Table 2). In exploratory analyses, the rates of some PBTs differ between genders. Survival is similar for male and females with similar NLR, Albumin, or ANC assays.

### A.L.A.N. Scores

An AS of 0 identifies the 34% of patients with a 54% (CI 41–68%) rate of ≥ 2-year survivors. The 47% of patients with an initial AS of 1–2, have an MST of 13 months, with a CI of 10.5–18 months, and 31% of the patients (CI 20–42%) survive ≥ 2 years. The HR of patients with an AS of 1–2 to AS 0 is 1.70 with a P value of 9.1x10$^{-3}$. For patients with an initial AS of 3–4, the MST is 3.8 months (CI 2.2–12 months). The survival HR of the 20% of patients with an AS of 3–4, to the patients with an AS 0–2 is 2.96. and the P value is 1.2x10$^{-6}$ (Fig 5A and 5B, Table 3).

The group with an AS of 0–2 has an MST of 17.9 months (CI 13.5–23) and 42 (CI 33–50) % survive > 2 years. A sensitivity analysis, the omission of patients with CCA, finds that patients

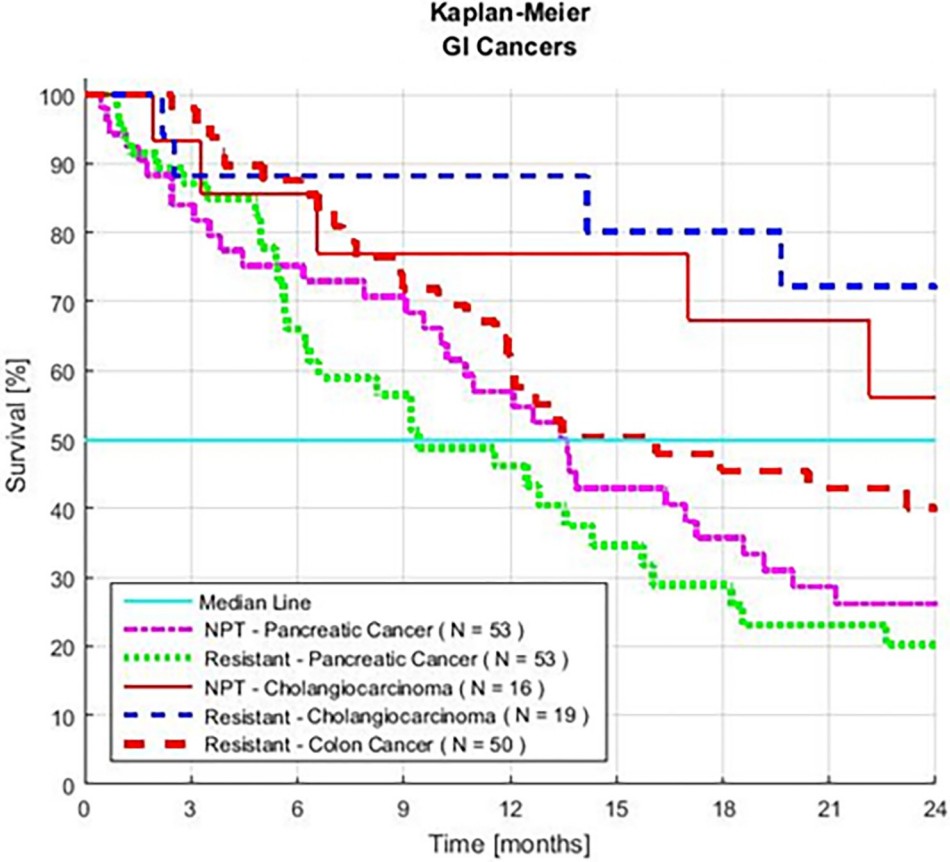

**Fig 1. Survival of five series of patients with advanced gastrointestinal cancer.** The patients were treated with GFLIO and no prior treatment (NPT) or prior treatment (resistant to) chemotherapy.

with an AS of 0–2 have an MST of 16 months and 35% (CI 25–45) survive > 2 years. When the tests described in this manuscript were examined in each disease, the qualitative information reproduced evidence that the same predictive tests and analyses apply.

### Prior treatment

The survivals in groups with prior therapy and no prior therapy appear to be similar (HR = 1.03, P = 0.88) (Fig 6, Table 2). There is a similar distribution of AS scores in both groups. Survival is similar when these patients are matched for an AS of 0 or 3–4. In exploratory analyses, the two groups with an AS of 1–2 have similar survival when the patients have similar serum albumin, NLR, or ANC (not shown).

### Individual tests

Patients have an MST of 11.6–13.9 months in groups defined by single PBTs that historically predict short survival: high NLR, platelets counts, and alkaline phosphates, and low LMR or lymphocytes. In contrast, patients with low albumin, high ANC, and low hemoglobin have the predicted short MSTs of 6.1, 3.9, and 5.6 months, respectively (Table 4).

Groups of patients defined by eight individual PBTs that predict long survival show a range of 37–54% chance of $\geq$ 2-year survival. For the 75% of patients with a serum albumin of $\geq$ 3.5 g/dL compared to those with a low count, the HR is 0.35 with a P value of $2.0 \times 10^{-7}$, the MST

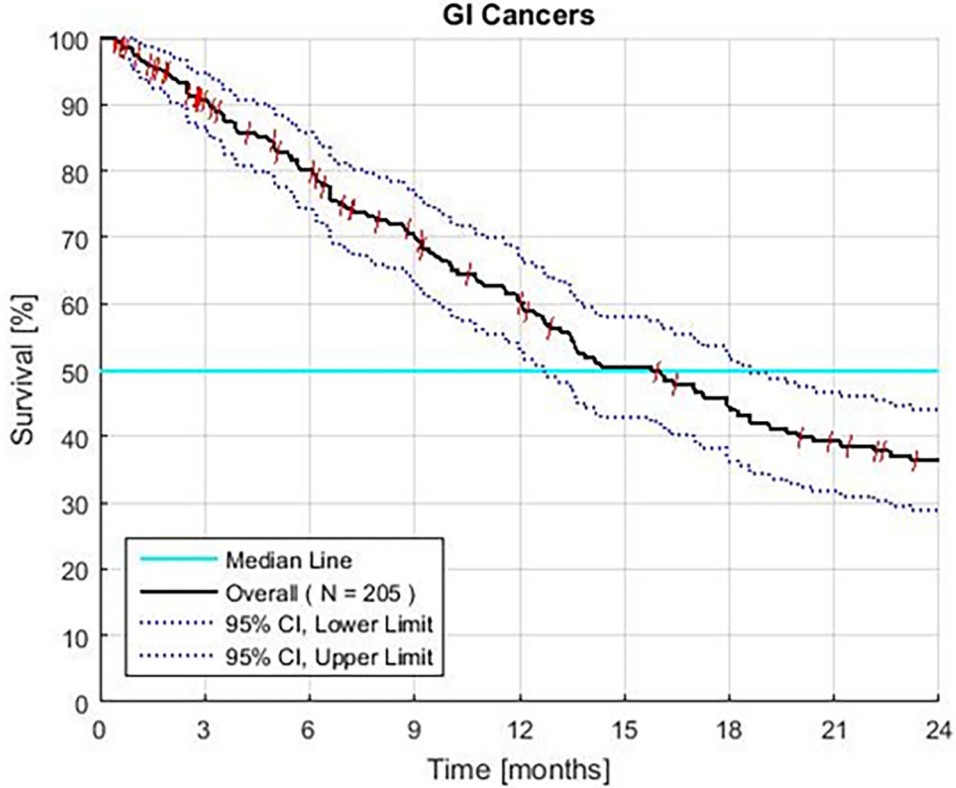

**Fig 2. Survival of all patients with advanced gastrointestinal (GI) cancer treated with GFLIO.** Third line colon, and both pancreatic, and intrahepatic bile duct tumors, with resistant and without prior treatment.

is 18.2 months, and 44% of patients survive $\geq$ 2 years. The 38% of patients with an NLR $<$ 3.0, have an HR and P value of 0.45 and $2.5 \times 10^{-5}$, respectively, and 54% of the group survives $\geq$ 2 years.

The 75% of patients with an ANC of $<$ 8000/μl, compared to the others (those with an ANC of $\geq$ 8000/μl, high count), have an HR and P value of 0.40 and $3.9 \times 10^{-5}$, respectively. Their MST is 17.2 months, and 43% of the patients survive $\geq$ 2 years. The 67% of patients with a platelet count of $\leq$ 300,000/μl, compared to the others (those with a high count), have an HR of 0.56, and P value of $1.8 \times 10^{-3}$. The MST of the groups are 18.6 and 12 months, respectively, and 44 vs. 21% of the patients survive $\geq$ 2 years (Table 4).

The LMR failed to be a statistically significant test overall; however, the group with an unfavorable LMR has an MST of 12 months, and there are large differences in the MSTs and 2-year rates of survival between groups with high and low LMRs (Table 4).

## Safety

There are neither novel nor limiting AEs (new side effects, nor side effects that prevent or delay treatment for $>$ 7 days), nor clinically significant change in the rates of grade 1–3 hematologic and allergic AEs compared to the rates observed during treatment with standard regimens [4, 7, 9, 12]. Real-time monitors did not find grade 3 enteritis, grade 3 neuropathy, neutropenic fever, nor hospitalizations due to chemotherapy. Hypothetically, drugs independent of dosage can produce idiosyncratic reactions and the rate of allergic reactions can increase with the length of treatment.

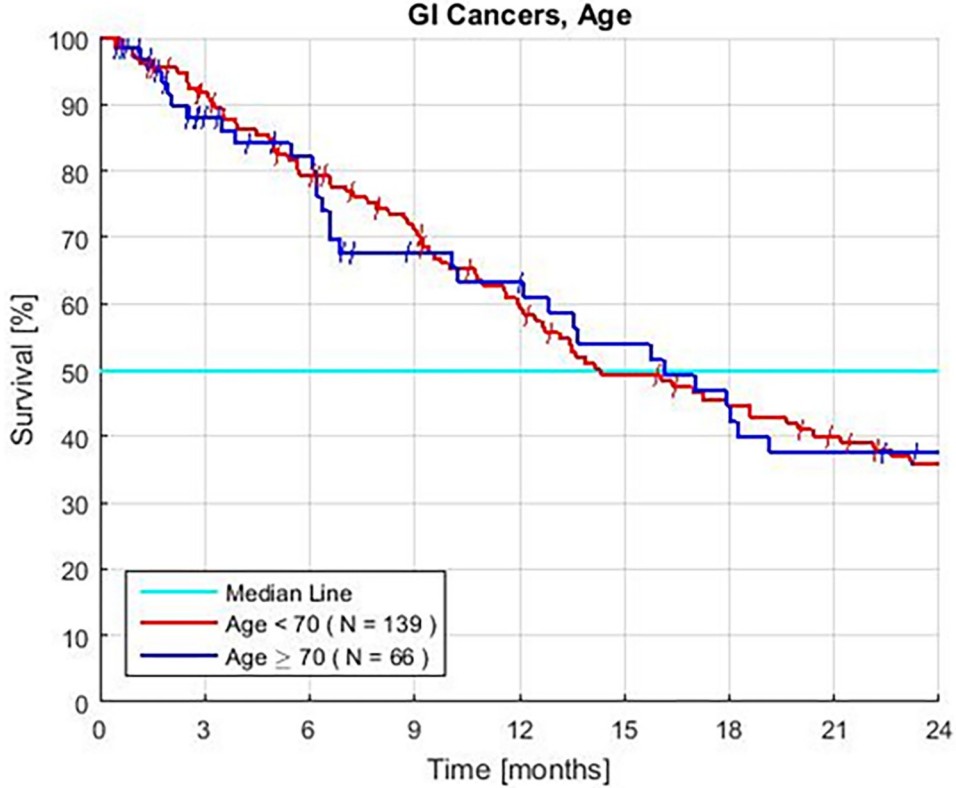

**Fig 3. Kaplan Meier survival of advanced gastrointestinal cancer patients separated by age.** Patients treated with GFLIO including resistant colon, pancreatic, and intrahepatic bile duct tumors, and no prior treatment (P = 0.77).

## Discussion

Real–world patients have an MST of 14.5 months. MSTs are reproducible [1]. The 81% of patients with an AS of 0–2, have an MST of 17.9 (CI 13.5–23) months and 42 (CI 33–50) % of those patients survive ≥ 24 months. Eight individual PBTs predict MSTs of ≥ 16 months. Five tests that predict short survival now identify groups of patients with an unexpected useful and possibly novel long MST of 11.6–13.9 months.

**Table 2. Patient characteristics and survival.**

| Characteristic: | Number of Patients | Percent of Patients | Median Survival | 2-year Survival | Hazard Ratio | HR ± 95 CI | P Value |
|---|---|---|---|---|---|---|---|
| – | # | % | Months | % | – | – | – |
| **ALL:** | 205 | 100 | 14.5 | 33 | – | – | – |
| Male | 110 | 54 | 13.1 | 32 | 1.41 | 0.99–2.00 | 0.05 |
| Female | 95 | 46 | 17.9 | 42 | 0.71 | | |
| < 70 years | 139 | 68 | 14.3 | 36 | 1.06 | 0.72–1.57 | 0.77 |
| ≥ 70 years | 66 | 32 | 16.1 | 38 | 0.94 | | |
| Prior Treatment | 122 | 60 | 14.2 | 37 | 1.03 | 0.72–1.46 | 0.88 |
| No Prior Treatment | 83 | 40 | 17 | 36 | 0.97 | | |

Advanced Cancers: 3rd Line Colon, Pancreatic, and both Intrahepatic Bile Duct Cancers with and without prior therapy

*60 ± 8% at 12 months.

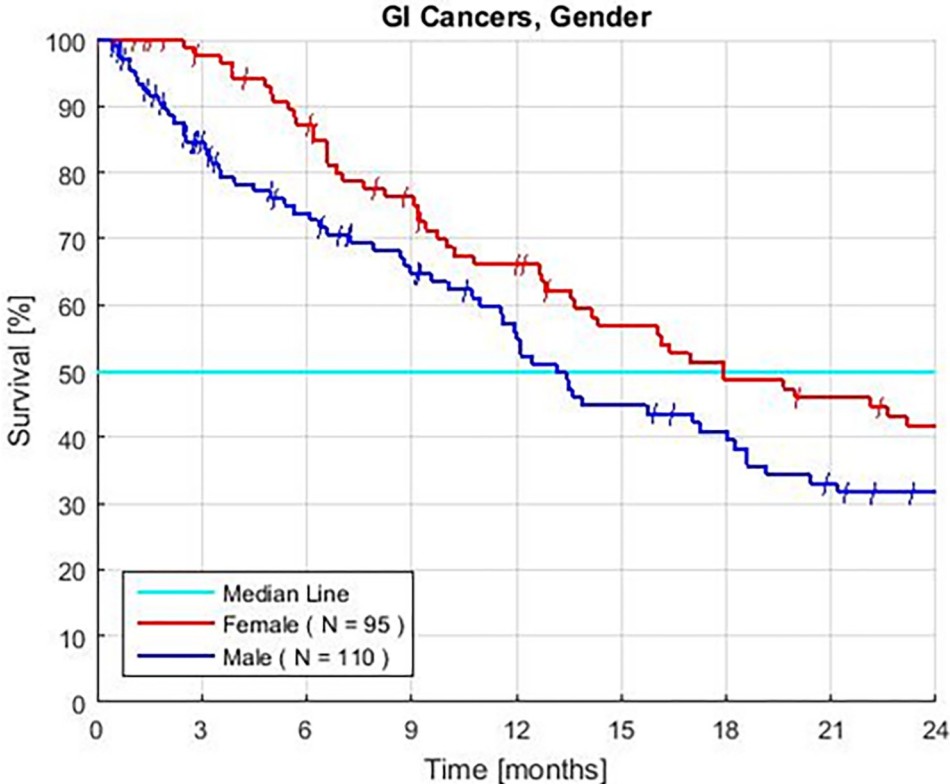

**Fig 4. Female vs. Male Kaplan—Meier survival of patients with advanced gastrointestinal cancers treated with GFLIO.** The overall P value = 0.05. The P value most strongly favors females with advanced pancreatic cancers.

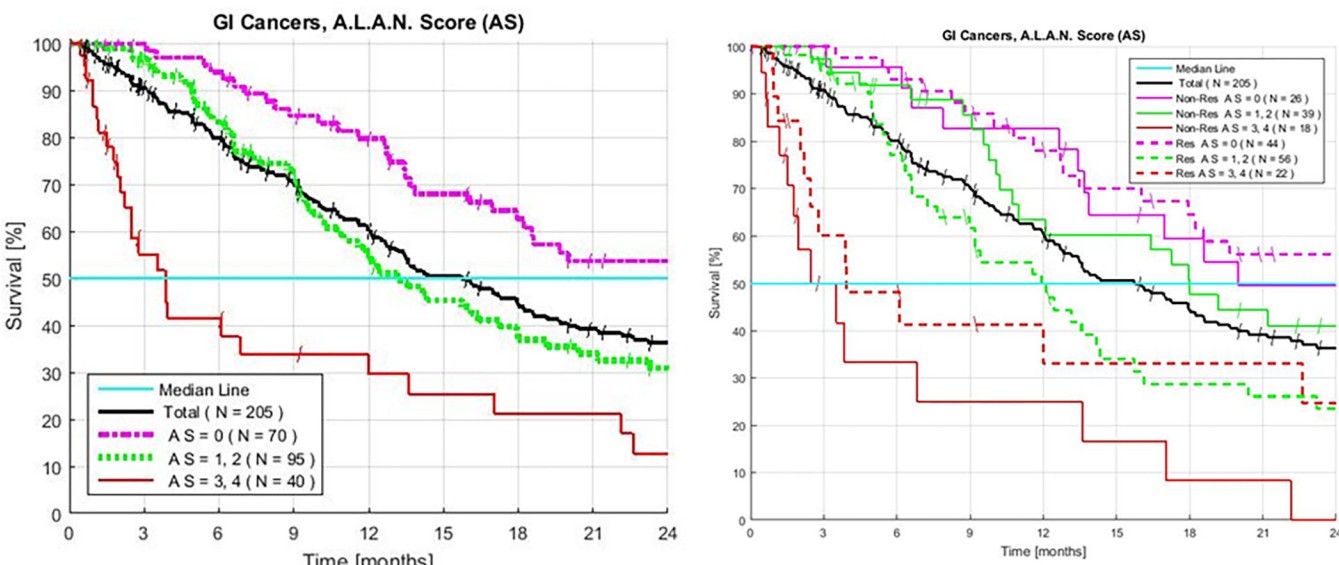

**Fig 5. A.** Overall combined Kaplan-Meier survival of patients with advanced gastrointestinal cancers treated with GFLIO. Resistant colon, and both pancreatic, intrahepatic bile duct tumors with and without prior treatment (AS 0 vs. 1–2, P < 9.1x10⁻³, HR 1.70. AS 0–2 vs. 3–4, P < 1.2x10⁻⁶, HR 2.69). **B.** Survival and AS 0, 1–2 or 3–4, tests (< 3.5 g/dL Albumin; > 3.0 NLR; > 8,000/mm³ neutrophils and a < 2.1 LMR), of patients with advanced gastrointestinal cancer treated with GFLIO. Resistant colon, and both pancreatic and intrahepatic bile duct tumors with and without prior treatment.

**Table 3. Combined A.L.A.N. score GI cancers and survival.**

| A.L.A.N Score | Number of Patients | Patients | Median Survival | 2-year Survival | Hazard Ratio | HR ± 95% CI | P Value |
|---|---|---|---|---|---|---|---|
| # | # | % | Months | % | – | – | – |
| 0–2 | 165 | 81 | 17.9 | 42 | 2.96 | 1.91–4.58 | $1.2 \times 10^{-6}$ |
| 3–4 | 40 | 19 | 3.8 | 13 | 2.96 | | |
| 0–1 | 134 | 65 | 18 | 43 | 2.13 | 1.47–3.09 | $6.2 \times 10^{-5}$ |
| 2–4 | 71 | 35 | 9.4 | 21 | 2.13 | | |
| 0 | 70 | 34 | >24 | 54 | 1.97 | 1.36–2.87 | $3.9 \times 10^{-4}$ |
| 1–4 | 135 | 66 | 11.9 | 26 | 1.97 | | |
| 1–2 | 95 | 47 | 13 | 31 | 1.7 | 1.14–2.53 | $9.1 \times 10^{-3}$ |

All patients treated with GFLIO for GI, advanced resistant colon, and both pancreatic, intrahepatic bile duct, with or without prior treatment. The P vs. all groups with higher A.L.A.N. Scores: 0, 1–2 or 3–4 tests: < 3.5 g/dL serum albumin; > 3.0 NLR; > 8,000/mm$^3$ neutrophils and a < 2.1 LMR [27].

The same pattern is observed with the PBTs for 81% of the patients with RAPC. Groups with > 12-month median survival include the 8 individual PBTs shown in Table 4, and an AS of 0 or 1–2. The 3 tests that historically predict very short, < 6-month median survival now have CI's that span 12-month survival rates. These include: platelets, alkaline phosphatase, and ANC. Patients have similar rates of each PBT compared to patients in registration trials [27, 29]. PBTs again identify groups of long and short survivors; therefore, tests and treatment may

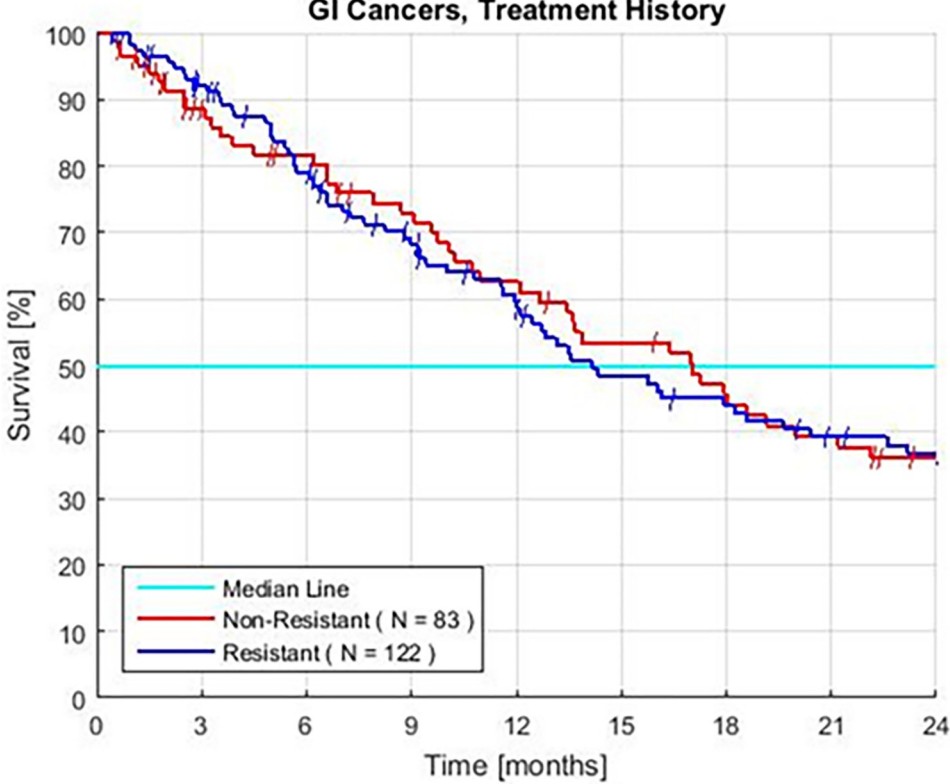

**Fig 6. Kaplan–Meier survival of patients displaying treatment history for patients with (resistant) and without (non-resistant) prior standard chemotherapy.** Resistant colon, and both pancreatic and intrahepatic bile duct tumors with and without prior treatment.

**Table 4. Combined GI cancers single prognostic characteristics and survival.**

| Test | Assay Value | Patients | Patients | Median Survival | 2-year Survival | Hazard Ratio | HR ± 95% CI | P Value |
|---|---|---|---|---|---|---|---|---|
| – | – | # | % | Months | % | – | – | – |
| Albumin | $\geq 3.5$ | 154 | 75 | 18.2 | 44 | 0.35 | – | $2.0 \times 10^{-7}$ |
| | $< 3.5$ | 51 | 25 | 6.1 | 10 | 2.89 | 1.94–4.31 | |
| Neutrophil-Lymphocyte Ratio | $< 3$ | 79 | 39 | $> 24$ | 54 | 0.45 | – | $3.1 \times 10^{-5}$ |
| | $\geq 3$ | 126 | 61 | 11.6 | 24 | 2.21 | 1.52–3.21 | |
| Absolute Neutrophil Count | $\leq 8$ | 166 | 81 | 17.2 | 40 | 0.40 | – | $3.9 \times 10^{-5}$ |
| | $> 8$ | 39 | 19 | 3.9 | 17 | 2.53 | 1.62–3.93 | |
| Platelets | $\leq 300$ | 138 | 67 | 18.6 | 44 | 0.56 | – | $1.8 \times 10^{-3}$ |
| | $> 300$ | 67 | 33 | 12 | 21 | 1.80 | 1.24–2.59 | |
| Alkaline Phosphate | $< 135$ | 76 | 37 | 18.6 | 47 | 0.63 | – | 0.02 |
| | $\geq 135$ | 129 | 63 | 12.1 | 30 | 1.58 | 1.09–2.28 | |
| Lymphocytes | $\geq 1.5$ | 113 | 55 | 17.9 | 41 | 0.69 | – | 0.04 |
| | $< 1.5$ | 92 | 45 | 12.1 | 30 | 1.45 | 1.02–2.06 | |
| Hemoglobin | $\geq 9$ | 193 | 94 | 16 | 38 | 0.50 | – | 0.06 |
| | $< 9$ | 12 | 6 | 5.6 | 14 | 1.99 | 0.99–4.07 | |
| Lymphocyte- Monocyte Ratio | $\geq 2.1$ | 165 | 81 | 16.1 | 37 | 0.78 | – | 0.28 |
| | $< 2.1$ | 40 | 19 | 12 | 35 | 1.29 | 0.82–2.02 | |

All patients treated with GFLIO for GI, resistant colon, and both pancreatic, and intrahepatic bile duct, with or without prior treatment.

be applicable to analyses of trials for well-defined homogenous patients with individual cancers across the spectrum of epithelial GI tumors [3, 23–25, 27–29].

Statistically strong tests include the AS 0–1, NLR, ANC, and platelet counts. Analyses of groups of as few as 30 patients can sometimes identify significant differences in survival; therefore, PBTs may facilitate personalized investigations designed to preempt, avoid, or correct mechanisms of lethality. These mechanisms include inflammation, cytokines, growth factors, and immunosuppression [23–28]. Absence of the pathophysiology associated with "unfavorable" PBTs may mitigate the expected lethal impact of prior therapy and age because the patients' survival can sometimes be similar when the PBTs are similar. Investigation of their impact requires large series and a meta-analysis, homogenous prior treatment, and inclusion of clinical characteristics such as prior responses, survival, lines of treatment, tumor burden, and PS.

Evaluation of the PBTs may assist phase 3 trials to investigate: long survival, sequential therapy, GFLIO in comparison to its component steps and to other regimens, safety, novel dosages, time to introduce either DM, or target drugs and the strategies incorporated in the algorithm. The latter includes re-challenge to reverse the resistance of tumors with recombind and supplementary new drugs; safety with novel, severely reduced dosages; combinations consisting of $\geq 4$ synergistic drugs compared to 1–2 drugs; and development of the regimen with additional drugs that satisfy novel laboratory criteria [10, 18, 32–36].

Objectives of new trials may include expanded criteria for eligibility, new methods of stratification and analyses, new cut-offs, and models that include tests of cellular and organ function. Tests may be surrogate measures of mechanisms of lethality.

In meta-analyses, PBTs can refine predictions of survival based on classic clinical prognostic characteristics such as; PS, stage, tumor volume, number and sites of metastasis, tumor marker assays, and the tumors' rate of growth. PBTs may represent a composite summary measure of life-threatening clinical characteristics [26, 27]. To our knowledge, no combination

of standard treatment and clinical prognostic criteria has identified groups of patients with the survival of the AS 0 and individual PBT groups in either the combined or individual series. Tests of eligibility can now include problematic patients when planned analyses and the utilization of PBTs serve as quality control tests [21].

Weak tests may facilitate the analyses of large trials and function as part of a model. Tests may only appear weak because of the unusually long MSTs of the group of patients with tests that have historically predicted intermediate and short survival. Hypothetically, a long MST compared to the literature can signal the treatments' impact on survival and lead to identification of mechanisms of benefit [27].

The use of PBTs may increase in importance in parallel with the need to evaluate new phase 2 regimens that have produced MSTs of about 18 months for selected patients with Karnofsky scores of 100–80. These new phase 2 trials also investigate individual strategies which are integrated in GFLIO: re-challenge, ± immunotherapy, the addition of taxanes, cetuximab, or bevacizumab, gemcitabine, and irinotecan [15, 16, 32–35, 37–39].

GFLIO-D is composed of the lowest dosages of these drugs that have been tested in empirical combination chemotherapy to date [40]. Use of 25–33% of standard dosages allows trials of combinations with 4–6 drugs to avoid all but 2–3% rates of severe AEs other than clinically silent nadir ANC and platelet counts. These dosages can slowly produce severe nadir blood counts. The warning time and the use of 1–2 injections of G-CSF can avoid delays of treatment [1, 2]. Stop-go practices, and when needed, delay in the introduction of a drug adds to the safety of the treatments.

Many mechanisms may contribute to survival in addition to an increase in the number of simultaneous interactions of drugs. These include: immunotherapy, metronomic effects, collateral sensitization, and the additive effects of > 4 simultaneous synergistic drug interactions. Clinical signals that GFLIO may improve the patients' immune system include complete responses and 5-year survivors, in series of patients with advanced gastric and intrahepatic bile duct cancers [41–44]. GFLIO ± DM can provide a safe backbone to test the simultaneous use of multiple potentiators of immunotherapy because the individual drugs, components of GFLIO, can be synergistic with immunotherapy [15, 16, 42–45].

Anecdotally, ad hoc re-challenge with GFLIO-DM after it failed may reverse resistance of CCAs to check point immunotherapy in the absence of PDL-1. Also, after GLFIO-DM fails or responses plateau, the intermittent addition of a PARP inhibitor to further treatment with GFLIO-DM can decrease the size of RAPC and CCA tumors, even in the absence of BRCA-like or HRD mutations.

A worsening of serial tests such as NLR may be early signals that treatment is ineffective [26]. Current analyses also support similar serial tests of changes in AS 0–1, albumin, and platelet counts to achieve timely change of treatment. Responses may also re-condition patients and produce opportunities for further investigation of intervention with regional therapy, elective sequential treatment, and chemoimmunotherapy to consolidate responses [2, 15, 16, 44, 45].

The decision to prioritize the development of safe sequential therapy and interactions of drugs precludes simultaneous tests of maximum tolerated dosages of the drugs. Prospective plans do not re-examine rates of: non-limiting gr 1–3 AEs, response to sequential treatment, nor the known interactions between PBTs and clinical prognostic characteristics [23–28]. Real-time monitors found that the safety and rates of response continue to satisfy criteria for further development of the sequential treatments [1, 2, 37, 41, 46–49]. Exploratory analyses found signal responses; neither PS > 40, nor 19–9 tumor marker levels in the thousands contraindicate treatment with GFLIO [1, 3, 41, 48, 49].

Caution requires the avoidance of untested practices including further reduction in dosage, current dosages in other combinations, or the use of GFLIO as a curative treatment. Some individual drugs fail when added to primary therapy. In contrast, as used in the GFLIO sequence, the drugs have new or many additional synergistic partners that are untested for the specific disease yet are already standard for other cancers. These include: irinotecan for cetuximab, docetaxel for bevacizumab, or a new application for the targeted drugs as a 3<sup>rd</sup> line treatment.

Further investigations require patients with individual cancers and homogenous prior therapy because the MSTs can vary by $>$ 12 months between series and between subgroups within each trial for a specific tumor. The PBTs may predict survival and serve as prospective selection criteria for patients of all ages, with or without prior treatment; however, the similar rates of many PBTs may be in part due to the unrecognized selection of patients in spite of efforts to include patients with severely abnormal clinical characteristics.

## Conclusion

PBTs and treatments have many features that encourage use and development in phase 3 trials. Objectives include: prolonged survival, safety, expansion of eligibility, and evaluation of several strategies to reverse the tumors' resistance to key drugs. The combination of treatment and tests may change practice for patients with many difficult cancers because of their unmet needs due to advanced age and poor tolerance or resistance to standard therapy. Aims include identification of well-defined groups of patients that can benefit from the regimens and identification of therapeutic hypotheses to avoid and correct the tests' underlying resistance mechanisms.

## Supporting information

**S1 Data.**
(PDF)

## Acknowledgments

All those who contributed to the work were listed as authors. We do not wish to add any acknowledgments.

## Author Contributions

**Conceptualization:** Howard W. Bruckner.

**Data curation:** Howard W. Bruckner, Fred Bassali, Elisheva Dusowitz, Abe Book.

**Formal analysis:** Howard W. Bruckner, Fred Bassali, Elisheva Dusowitz, Abe Book.

**Funding acquisition:** Howard W. Bruckner.

**Investigation:** Howard W. Bruckner.

**Methodology:** Howard W. Bruckner, Fred Bassali.

**Project administration:** Howard W. Bruckner.

**Resources:** Howard W. Bruckner.

**Software:** Fred Bassali, Daniel Gurell.

**Supervision:** Howard W. Bruckner.

**Validation:** Howard W. Bruckner, Daniel Gurell, Robert De Jager.

**Visualization:** Howard W. Bruckner, Daniel Gurell.

**Writing – original draft:** Howard W. Bruckner, Elisheva Dusowitz, Robert De Jager.

**Writing – review & editing:** Howard W. Bruckner, Fred Bassali, Elisheva Dusowitz, Abe Book, Robert De Jager.

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
