## [Decision Letter · Decision Letter 0]

19 Jul 2022

PONE-D-22-16131Actionable Tests and Treatments for Patients with Difficult Gastrointestinal CancersPLOS ONE

Dear Dr. Bruckner,

Thank you for submitting your manuscript to PLOS ONE. After careful consideration, we feel that it has merit but does not fully meet PLOS ONE’s publication criteria as it currently stands. Therefore, we invite you to submit a revised version of the manuscript that addresses the points raised during the review process.

We look forward to receiving your revised manuscript.

Kind regards,

Md Sazzad Hassan

Academic Editor

PLOS ONE

Journal Requirements:

Support: The Marcus foundation, MZB Foundation for Cancer Research, Aid L’Shalom Foundation

 Funding: The Marcus foundation, MZB Foundation for Cancer Research, Aid L’Shalom Foundation

3. PLOS requires an ORCID iD for the corresponding author in Editorial Manager on papers submitted after December 6th, 2016. Please ensure that you have an ORCID iD and that it is validated in Editorial Manager. To do this, go to ‘Update my Information’ (in the upper left-hand corner of the main menu), and click on the Fetch/Validate link next to the ORCID field. This will take you to the ORCID site and allow you to create a new iD or authenticate a pre-existing iD in Editorial Manager. Please see the following video for instructions on linking an ORCID iD to your Editorial Manager account: https://www.youtube.com/watch?v=_xcclfuvtxQ.

Reviewers' comments:

Reviewer's Responses to Questions

**Comments to the Author**

1. Is the manuscript technically sound, and do the data support the conclusions?

Reviewer #1: Yes

Reviewer #2: Yes

2. Has the statistical analysis been performed appropriately and rigorously? 

Reviewer #1: Yes

Reviewer #2: Yes

3. Have the authors made all data underlying the findings in their manuscript fully available?

Reviewer #1: Yes

Reviewer #2: Yes

4. Is the manuscript presented in an intelligible fashion and written in standard English?

Reviewer #1: Yes

Reviewer #2: No

5. Review Comments to the Author

Reviewer #1: This manuscript describes a possible treatment for patients with advanced gastrointestinal cancers. The authors then proceed to investigate/validate their results based on published predictive blood tests.

Combination of several drugs with different mechanism of actions is an important strategy for the treatment of these patients, as it might overcome resistance to single agents.

While there remains an undisputed treatment need for patients with advanced GI cancers, several concerns remain with the current manuscript that should be addressed.

1. Title: Unclear what the scope of the current manuscript is. What are difficult gastrointestinal cancers? Please clarify.

2. Abstract: Incoherent, please rephrase to more adequately reflect the study and its results.

Line 44: part of the sentence seems missing

3. Introduction: Large parts of the introduction should be in the Methods section. Consider adding background information on advanced gastrointestinal cancers, and the challenges these patients face. This would explain the rationale for this study. Better explain the A.L.A.N. score for readers that might not be familiar with it.

4. Methods: Is this a clinical trial? Please describe where the patients were treated and how they were selected? What were the reason to omit standard treatment in treatment naïve patients? Were they offered standard treatment? Explain NCI grade blood tests.

5. Results: Please define the different series better to avoid confusion.

Line 287: 37 - 54% chance to survive 2 years? Correct number.

6. Discussion: Real world patients with advanced GI cancers... Please explain the 13 subgroups for PAPC.

Line 320: 73-83% of patients with AS of 0-2?

Line 321: 41.6 patients?

Line 322: MST of >16 to > 24 month?

Table 1: All, correct 2 year survival rate and CI

In general, the authors should improve the flow of the narrative of this manuscript.

Reviewer #2: Line 28

“Patients have unmet needs when chemotherapy produces either a median survival of less than 1 year or severe toxicities. Blood tests can predict survival. Resistance to the drugs can be safely reversed with empirical re-combination of drugs and steep reduction of dosages, respectively.”

->Presumably this should read resistance to specific chemotherapeutic agents or severe toxicities can be mitigated by empirical recombination of agents or steep dose reductions, respectively.

Line 110

“The A.L.A.N. score (AS), the number of individual tests (serum albumin < 3.5 g/dL,

neutrophil/lymphocyte ratio (NLR) > 3.1, absolute neutrophils (ANC) > 8,000/mm3, and

lymphocyte/monocyte ratio (LMR) < 2.1) predict the survival chances of patients with primary CCA

->One of many sentences that are difficulty to understand. Are the authors stating to the A.L.A.N as well as the number of individual tests within that range separately predict survival? The second reference at the end of that sentence involve a retrospective trial of pancreatic cancer not cholangiocarcinoma. Perhaps better phrased as “The A.L.A.N. score is a prognostic scoring system employing readily available laboratory studies that stratifies patients into risk groups correlated with overall survival in patient undergoing first-line chemotherapy for cholangiocarcinoma [ref 4]. In this study, the authors employ a modified A.L.A.N score where one point each is given for each of the following criteria: baseline absolute neutrophil count > 8,000/mm3, lymphocytes-monocytes ratio (LMR) < 2.1, neutrophil-lymphocytes ratio (NLR) > 3.1, and albumin <3.5 g/dL”

->The A.L.A.N score cited in ref 4 used a NLR of 3 rather than 3.1 as the authors employ in this paper. What is the reasoning for this difference? Was it a typographic error? In Table 3 the typical criteria for this score are use, i.e. NLR of 3.

Line 116

RAPC is used without defining it

Line 214

Table 1 has entries of <70 and greater than or equal to 70 but does not state this is referring to age. Perhaps add the word “years”

The authors touch on it briefly, but given the difference in the Kaplan-Meier curves for cholangiocarcinoma (not spelled correctly in Figure 1 legend) versus pancreatic and colon cancer, how does the model do with each individual cancer or with each of these cancer types removed from the analysis?

Why is the survival of resistant A S 3,4 worse than non-resistant A S 3,4 if figure 5B, is this simply due to the small n’s?

Ultimately the paper is examining primarily the predictive value of the A.L.A.N. score in patients receiving GFLIO chemotherapy. It would seem to be a more straightforward title and thrust to the paper.

6. PLOS authors have the option to publish the peer review history of their article (what does this mean?). If published, this will include your full peer review and any attached files.

Reviewer #1: No

Reviewer #2: No

---

## [Author Response · Author response to Decision Letter 0]

19 Aug 2022

Response to Journal Requirements:

1. Our manuscript now meets PLOS ONE’s style requirements including those for file naming.

2. We removed funding information from the acknowledgements section of our manuscript and approve of it being written as follows: 

Funding: The Marcus Foundation, MZB Foundation for Cancer Research, and Aid L’Shalom Foundation. The funders had no role in study design, data collection and analysis, decision to publish, or preparation of the manuscript, with the exception of Dr. Howard Bruckner as the Chief Science Officer of the MZB Foundation. 

3. The newly created ORCID for the corresponding author is up to date and entered in the Editorial Manager. Please see the attached link for your reference: Howard W Bruckner, HW Bruckner (0000-0002-7459-7255) (orcid.org)

4. A full ethics statement appears in our Methods Section and describes the consent as written. It was designed and approved by the IRBs. 

Reviewer #1 – Response to Editorial Comments:

1. The title of the paper clarifies the term by replacing “difficult” with “historically short median survival times.”

2. The abstract now reflects the study and its results. Line 44 in the previous manuscript was removed in the revised copy due to space limitations. 

3. The introduction is consolidated. Parts of the introduction that must be in the methods section were moved there to avoid duplication. The introduction now starts with the background of historical gastrointestinal survivals and adverse event experiences. Added information provides rates of American and worldwide patients that die per annum. These are the patients that have these unmet needs each year. The function and objectives of the A.L.A.N. Score are described in the introduction, in lines 114-125. The methods of the A.L.A.N. Score are now described in the methods section, in lines 165-169. 

4. This is a clinical trial conducted in an outpatient oncology clinic. The patients were treated in a single practice, coordinated with hospital IRBs. The trial was expanded with IRB approvals because of the compassionate unmet needs and additional objective of more information, identification of the characteristics of patients that will gain benefit, and therapeutic hypotheses for investigation. This information was placed in the Methods section, in lines 212-214.

The selection process occurred over a 2-year period, as stated in line 211. The cut off dates avoid the need to parse the impact of interruptions in referral and treatment due to the pandemic. 

Intent-to-treat analyses (select) evaluate the most difficult patients in order to demonstrate proof of principal, with the most difficult, stage-4 diagnoses, and patients with unmet needs. These are the series for which we have adequate numbers, 36-50 patients in each group, not anecdotes. The information regarding the prognostic blood tests is new and applicable to unmet needs for patients with resistant colorectal cancer and cholangiocarcinoma. Physicians need to know and be able to identify patients that can often benefit from treatment. 

Ineligibility is minimized because the objective is to provide treatment to patients with unmet needs. Our regimen has twenty years of safety due to low dosage. Exclusion ineligibility criteria prioritize safety, office care, and criteria that are potentially suitable for institutional cancer centers and cooperative groups. This was a deliberate effort to minimize exclusion in order to test and cautiously expand limits of eligibility. Prior experience with this project evolved in step wise fashion due to an ever-increasing volume of referrals for “untreatable, last resort” patients. We were pressured to continue to test the limits of safety and eligibility because safety allows treatment and the benefit rate of > 50% encouraged continuation. 

The reason to omit standard practice is because independent expert consultants advised against it because they anticipated median survival times of < 10 months due to diagnoses stage, volume of disease, and complications. This was added into the manuscript and explained in the methods section, lines 176-178. 

The practice adheres to NCI grade blood tests as used by the cooperative groups because they make sense and avoid a level of complexity that would preclude adoption by other investigators. This was explained in the Methods section, in lines 171-172. The grade 0-2 blood tests are defined in reference 30. 

5. The different series are clearly explained. Line 287 in the previous manuscript which is Line 278 in the revised copy now reads the correct numbers, including an HR of 2.96 and P value of 1.2x10-6. 

6. Discussion: individual tests that predict favorable survival are shown in Table 3. These include: high serum albumin, high NLR, low ANC, low platelets, low alkaline phosphates, high lymphocytes, high hemoglobin, and high LMR. The favorable test groups for resistant advanced pancreatic cancer patients are enumerated in the text. The discussion briefly mentioned it because they demonstrate the consistent potential utility of the prognostic blood tests (This is applying to more heavily treated groups, with applications to ~ 80% of patients that are striking novel demonstrations.) The tests that predict short MST are now included in line 374. 

• Line 320 in the previous manuscript, now line 366 in the revised copy: The 73-83% of patients with an AS of 0-2. This was simplified to 81%. 

• Line 321 in the previous manuscript, now line 367 in the revised copy: The 41.6 patients. This was simplified to the 42% of the patients with A.L.A.N. Scores of 0 – 2. 

• Line 322 in the previous manuscript, now line 368 in the revised copy: MSTs of ≥ 16 to ≥ 24 months. This was simplified to say ≥ 16 months. 

• Table 1: 2-year survival rate and CI for all the patients were corrected.

Reviewer #2 – Response to Editorial Comments: 

• Line 28 in the previous manuscript: Due to space limitations and to avoid repetition, this line was integrated into the conclusion in lines 48-49.

• Line 110 in the previous manuscript: The A.L.A.N. Score consists of the number, 0-4, of individual unfavorable tests. This was clarified by having the paper read “and also individual tests” in line 114. The correct reference was also placed in the revised manuscript. We encorporated the reviewer’s A.L.A.N. score clarification into our introduction, in lines 114-125. We explain the model, how it is scored, the disease Cholangiocarcinoma, and the illustrative result. 

• NLR has the value of 3.0, now corrected in the manuscript. The reference at the end of the sentence, line 118, is now corrected (reference 29). 

• Line 116 in the previous manuscript, line 121-122 in the revised copy: RAPC’s preferred abbreviation is inserted.

• Table 1: The word “years” is added to clarify that < 70 and ≥ 70 refers to age. 

• The spelling of cholangiocarcinoma was corrected in the Figure 1 legend.

• In the results section, lines 297-300, we describe a sensitivity analysis omitting cholangiocarcinoma. We also included a line explaining that the A.L.A.N. score and individual tests produced similar findings for each disease.

• The “observed” disparities between resistant and non-resistant A.L.A.N. Score 3-4 patients may be due to several factors. First, the small number of patients are not significant. Second, there may hypothetically be a selection with inadvertent referral differences. Patients with no prior treatment may have a lower threshold, worse AS 3-4, whereas A.L.A.N. Score 3-4 patients with prior treatment are less likely to be referred for another treatment. This is illustrated when the groups are matched for the most powerful individual tests; the difference disappears, but the numbers remain exploratory. This was not discussed in the manuscript unless it is the reviewers’ recommendation to do so. 

• Ultimately the paper is examining primarily the predictive value of the A.L.A.N. score in patients receiving GFLIO chemotherapy. It would seem to be a more straightforward title and thrust to the paper. We agree with this statement; however, we want to make the point of expanding the tests to patients with unmet needs due to short survival, treatment resistance, and age.

---

## [Decision Letter · Decision Letter 1]

10 Oct 2022

Actionable Tests and Treatments for Patients with Gastrointestinal Cancers and Historically Short Median Survival Times

PONE-D-22-16131R1

Dear Dr. Bruckner,

We’re pleased to inform you that your manuscript has been judged scientifically suitable for publication and will be formally accepted for publication once it meets all outstanding technical requirements.

Kind regards,

Md Sazzad Hassan

Academic Editor

PLOS ONE

Additional Editor Comments (optional):

Reviewers' comments:

Reviewer's Responses to Questions

**Comments to the Author**

1. If the authors have adequately addressed your comments raised in a previous round of review and you feel that this manuscript is now acceptable for publication, you may indicate that here to bypass the “Comments to the Author” section, enter your conflict of interest statement in the “Confidential to Editor” section, and submit your "Accept" recommendation.

Reviewer #1: All comments have been addressed

Reviewer #2: All comments have been addressed

2. Is the manuscript technically sound, and do the data support the conclusions?

Reviewer #1: Yes

Reviewer #2: Yes

3. Has the statistical analysis been performed appropriately and rigorously? 

Reviewer #1: Yes

Reviewer #2: Yes

4. Have the authors made all data underlying the findings in their manuscript fully available?

Reviewer #1: No

Reviewer #2: Yes

5. Is the manuscript presented in an intelligible fashion and written in standard English?

Reviewer #1: Yes

Reviewer #2: Yes

6. Review Comments to the Author

Reviewer #1: (No Response)

Reviewer #2: There remain some sentences that are somewhat awkwardly worded as in the abstract background, but they care comprehensible. The concerns brought up on the prior review have been adequately addressed.

7. PLOS authors have the option to publish the peer review history of their article (what does this mean?). If published, this will include your full peer review and any attached files.

Reviewer #1: No

Reviewer #2: No

---

## [Editor Report · Acceptance letter]

24 Oct 2022

PONE-D-22-16131R1 

Actionable Tests and Treatments for Patients with Gastrointestinal Cancers and Historically Short Median Survival Times 

Dear Dr. Bruckner:

I'm pleased to inform you that your manuscript has been deemed suitable for publication in PLOS ONE. Congratulations! Your manuscript is now with our production department. 

Kind regards, 

on behalf of

Dr. Md Sazzad Hassan 

Academic Editor

PLOS ONE